# Combined Effects of Obesity and Dyslipidaemia on the Prevalence of Diabetes Amongst Adults Aged ≥45 Years: Evidence from a Nationally Representative Cross-Sectional Study

**DOI:** 10.3390/ijerph19138036

**Published:** 2022-06-30

**Authors:** Simin Zhang, Donghan Sun, Xiaoyi Qian, Li Li, Wenwen Wu

**Affiliations:** 1School of Public Health, Hubei University of Medicine, Shiyan 442000, China; zhangsimin0266@163.com (S.Z.); qianxiaoyi5@163.com (X.Q.); 2Institute for Evidence-Based Nursing, Renmin Hospital, Hubei University of Medicine, Shiyan 442000, China; syryhlb_sdh@163.com

**Keywords:** body mass index, waist circumference, dyslipidaemia, combined effect, diabetes

## Abstract

Objectives: This study aimed to explore the combined effects of different types of obesity and dyslipidaemia on the prevalence of diabetes in middle-aged and elderly residents. Methods: Data were obtained from the 2015 China Health and Retirement Longitudinal Studydatabase, and 5023 valid participants were included after excluding those with missing data. A Chi-square test was used to test the difference in the prevalence of diabetes between the groups. Binary logistic regression was used to analyse the relationship between different types of obesity combined with dyslipidaemia and the prevalence of diabetes. Results: Multivariate logistic regression analysis showed that, compared with those with a body mass index <24/ortholiposis, the subgroup with systemic obesity/dyslipidaemia had 4.37 times the risk of diabetes (OR = 4.37, 95% CI = 2.36–8.10, *p* < 0.001). In addition, compared with those with a normal waist circumference (WC)/ortholiposis, the subgroup with abdominal obesity/dyslipidaemia had 3.58 times the risk of diabetes (OR = 3.58, 95% CI = 2.49–5.13, *p* < 0.001). Conclusions: The coexistence of obesity and dyslipidaemia can significantly increase the risk of diabetes, suggesting that the strict control of weight, WC and lipid level is beneficial to the prevention of diabetes.

## 1. Introduction

Global populations around are rapidly aging [1]. The aging of global populations is the most important medical and socio-demographic problem in the world. By 2050, the proportion of the population aged 65 years and over will reach 38% in several countries [2]. Age growth occurs the fastest in developing countries, especially in China [3]. In 2016, the Chinese population aged 60 years and over was over 230 million, and it will further increase to 480 million by 2050 [4].

Diabetes is a kind of endocrine metabolic disease characterised by hyperglycaemia. According to the diabetes map of the International Diabetes Federation, 451 million patients with diabetes were estimated in the world in 2017 [5]. Elderly people are at a high risk of diabetes [6]. By 2045, the prevalence of diabetes worldwide will rise to 10.9% (700 million people) [7]. In China, the prevalence of diabetes was 0.67% in 1980, and it increased to 10.9% and 11.2% in 2013 and 2015, respectively [8,9]. The prevalence of diabetes is increasing rapidly. In China, diabetes is most common amongst adults aged 45 years or above, but the treatment control rate is low [10,11].

Diabetes plays a major role in the numerous factors that contribute to cardiovascular disease (CVD), and people with diabetes are at a high risk for CVD [12]. The risk of cardiovascular events in females with diabetes is 25–50% higher than that in males [13]. The morbidity and mortality rates from CVD remain high in China, with an estimated 4 million people dying from CVD each year [14]. In addition, CVD was the leading cause of death in a study of causes of death in urban and rural residents [15]. One of the key measures to reducing the prevalence of CVDs is to reduce the prevalence of diabetes in the population [16]. Previous studies on risk factors for CVD showed overweightness and obesity as the main factors [17]. Waist circumference (WC) and body mass index (BMI) are the common indicators of obesity, and they have a good screening value for diabetes [18].

Dyslipidaemia can lead to various chronic CVDs [19], such as diabetes [20]. Diabetes mellitus type 2 is a metabolic disease characterised by hyperglycaemia and consisting of high glucose levels chronically present in the blood. Oxidative stress is critically involved in diabetes pathogenesis. The important role of red cell sensitivity to oxidative stress in different physiological conditions has been extensively studied. Glycosylated haemoglobin (A1c) is a standard test for the monitoring of blood sugar status [21]. The higher the blood lipid level, the higher the blood glucose level; in addition, dyslipidaemia can increase the risk of diabetes [22]. Observations confirmed the putative utility of dyslipidaemia in CVD risk prediction in individuals with diabetes [23]. Elevated low-density lipoprotein cholesterol (LDL-C) is an established risk factor that may increase the risk of diabetes [24,25]. Obesity is the most common nutritional disorder worldwide, and it is one of the major risk factors for atherosclerotic CVD [26]. Specifically, low high-density lipoprotein cholesterol (HDL-C) shows the strongest correlation with obesity. Obesity itself is not the cause of dyslipidaemia [27]. As suggested by a current review, the main drivers of dyslipidaemia comprise ectopic lipid deposition, lipid dysmetabolism and adipose tissue dysfunction [28]. Although the current studies have reported the independent effects of obesity and dyslipidaemia on the prevalence of diabetes, few researchers have studied the combined effects of the two factors. The present study used the data from the China Health and Retirement Longitudinal Study (CHARLS) in 2015 to analyse the relationship amongst BMI, WC, dyslipidaemia and the prevalence of diabetes in the middle-aged and elderly Chinese populations and provide a reference for the formulation of primary prevention strategies for diabetes.

## 2. Materials and Methods

### 2.1. Design

CHARLS is a national representative survey of Chinese adults aged 45 years and over [29], and it was first carried out by Peking University in 2011. A new round of follow-up surveys is launched every 2 years. CHARLS uses questions similar to those of the Health and Retirement Survey in the US and provides extensive demographic and socioeconomic statuses of health data. The present study was based on the cross-sectional data of the 2015 CHARLS. Face-to-face interviews in the respondents’ homes were conducted to collect detailed information on their demographic characteristics, socioeconomic status, health-related behaviour and lifestyles [30]. CHARLS utilises a multi-stage probability-proportional-to-size sampling technique, wherein participants were selected from 450 villages in 150 counties with 28 provinces, autonomous regions and municipalities directly under the central government [29]. The investigators randomly selected sample households from the list of all households in the village/neighbourhood committee of each sample [29]. Informed consent was obtained from the sample households before the survey. The inclusion criteria were as follows: (1) age ≥ 45 years, (2) living at the survey site for at least 6 months and (3) the ability to participate in the study during the investigation period. Individuals with mental retardation and severe cognitive impairment were excluded from the initial research. Only those requiring minimal or no help to answer the questions were allowed to participate in the study. As shown in Figure 1, after missing data samples and samples of patients with other CVDs and cancer had been excluded, the data of 5023 participants were included for further studies. The data collection was approved by the Biomedical Ethics Review Committee of Peking University (IRB00001052–11015) [31]. Further details on the CHARLS survey design are available from the work of Zhao et al. [29].

### 2.2. Assessment and Measurements

#### 2.2.1. Sociodemographic Characteristics

The questionnaire was used to collect detailed information, including age (year), gender (female or male), marital status, educational level, residence, exercise, smoking, alcohol drinking, nap duration and night sleep duration. People’s ages were divided into three groups (≤60, 61–70 and ≥71); education levels were divided into three categories (illiterate, primary school/middle school and high school and above); and marital statuses were divided into two categories, namely, (1) married with spouse present or married but not living with spouse temporarily for reasons such as work, separation or cohabitation, and (2) divorced, widowed or never married.

#### 2.2.2. Measurement of Haemoglobin A (HbA1c) and Blood Lipid Levels

All blood samples were stored at −80 °C at the China Centre of Disease Control and Prevention and tested in the clinical medicine laboratory of Capital Medical University. Glycosylated haemoglobin, HDL-C, LDL-C, total cholesterol (TC) and triglyceride (TG) were measured using enzyme colorimetry [29].

#### 2.2.3. Health Status

In this study, health status was assessed by ‘self-reported’ and reference definitions.

In accordance with the American Diabetes Association criteria, diabetes was defined as: fasting plasma glucose ≥7.0 mmol/L and/or glycosylated HbA1cconcentration of 6.5% or higher, taking diabetes medications [32] or self-reported doctor diagnosis of diabetes if they answered ‘yes’ to the question: ‘Have you been diagnosed with diabetesor high blood sugar?’

Dyslipidaemia was defined as TC < 6.22 mmol/L, LDL-C < 4.14 mmol/L, HDL-C ≥ 1.04 mmol/L or TG < 2.26 mmol/L or current treatment with lipid-lowering medications [33]. Dyslipidaemia was determined using the question, ‘Have you been diagnosed with dyslipidaemia?’ with an answer of ‘Yes’ or ‘No’.

#### 2.2.4. Sleep Duration

The sleep duration in the study included night sleep and naps and was recorded by asking the following questions: (1) ‘During the past month, how long did you take a nap after lunch?’ and (2) ‘During the past month, how many hours of actual sleep did you get at night (average hours for one night)? (this may be shorter than the number of hours you spend in bed).’ Naps are a common behaviour in human life. The longer the nap duration in the daytime (≥90 min), the higher the prevalence of metabolic syndrome [34]. Nap duration (minutes) was divided into three groups: 0, 0–30 and ≥30. According to the international clinical diagnostic standards, night sleep duration (h) has two categories: ≤5 and >5 [35].

#### 2.2.5. Physical Measurement

Physical parameters, such as the respondents’ standing height, weight and WC, were measured by trained investigators with standardised equipment (index: height, SecaTM213 Stadiometer, China Seca, Co., Ltd. (Hangzhou, China); index: weight, OmronTMHN-286 Scale, Krill Technology, Co., Ltd. (Yangzhou, China); index: WC, soft tape measure). BMI was calculated as follows: weight (kg)/height (m)^2^. The participants in this study were classified as underweight (<18.5 kg/m^2^), normal weight (18.5–23.9 kg/m^2^), overweight (24.0–27.9 kg/m^2^) or obese (≥28.0 kg/m^2^). Abdominal obesity was defined as a WC of 85 cm or greater for males and 80 cm or greater for females [36].

### 2.3. Statistical Analysis

All data analysis and tests were conducted using Stata (version 16.0, StataCorp, College Station, TX, USA). Categorical variables were expressed using the Chi-square test. Continuous variables were presented as the means ± standard deviation. The relationship amongst different obesity types, dyslipidaemia and the prevalence of diabetes was plotted using SigmaPlot12.0. The association amongst BMI, WC, dyslipidaemia and diabetes (odds ratios (ORs) and 95% confidence intervals (CIs)) was calculated by multivariate logistic regression. Statistical significance was set at *p* < 0.05.

## 3. Results

### 3.1. Subject Characteristics

Table 1 indicates that the age range of the 5023 participants included in the analysis was 45–94 years, with an average age of 57.85 ± 9.75 years. Amongst them, 1202 (23.93%) were male, with an average age of 56.97 ± 10.66 years. More than half of the participants (76.07%) were female, with an average age of 58.12 ± 9.43 years. BMI was classified into three groups (<24, 24–27.9 and ≥28 kg/m^2^), with a corresponding prevalence of diabetes of 42.10%, 38.74% and 19.16%, respectively. In the abdominal obesity group (yes or no), the constituent ratios of diabetes were 23.92% and 76.08%, respectively. In the abnormal blood lipid group, the prevalence rates of diabetes were 91.47% and 8.53%, respectively. In addition, *p* < 0.05 reflected statistically significant differences amongst these independent variables in diabetes.

### 3.2. Relationships between Obesity/Dyslipidaemia and Diabetes

Table 2 shows the results of the multivariate logistic regression analyses. After adjusting for age, hypertension, smoking status, abdominal obesity and dyslipidaemia, the risk of diabetes in the group with systemic obesity was higher than that in the group with non-systemic obesity (OR = 1.56, 95% CI = 1.28–1.89). The risk of diabetes in abdominal obesity was 1.92 times higher than that in the subgroup with non-abdominal obesity. The participants with dyslipidaemia had a 1.82 times higher risk of developing diabetes than those with normal blood lipids (*p* < 0.05).

### 3.3. Combined Effects of Systemic-Type Obesity and Dyslipidaemia on Diabetes

As illustrated in Figure 2, the subjects were divided into six groups based on the data on blood lipids and BMI. The group with a BMI < 24 kg/m^2^ combined with dyslipidaemia, a BMI of 24–27.9 kg/m^2^ combined with dyslipidaemia and a BMI ≥ 28 kg/m^2^ combined with dyslipidaemia had a higher prevalence of diabetes. The difference in prevalence amongst the six groups was statistically significant (χ^2^ = 66.74, *p* < 0.001).

As shown in Figure 3, the multivariate logistic regression analysis showed that, compared with the group with a BMI < 24/ortholiposis, the risks of diabetes in patients with a BMI < 24 combined with dyslipidaemia, a BMI of 24–27.9 combined with dyslipidaemia and a BMI ≥28 combined with dyslipidaemia were 1.98 times (OR = 1.98, 95% CI = 1.15–3.41, *p* = 0.014), 3.10 times (OR = 3.10, 95% CI = 1.97–4.85, *p* < 0.001) and 4.37 times (OR = 4.37, 95% CI = 2.36–8.10, *p* < 0.001) higher, respectively.All values were higher compared with those of the group with a BMI < 24/ortholiposis.

### 3.4. Combined Effects of Abdominal Obesity and Dyslipidaemia on Diabetes

As illustrated in Figure 4, the subjects were divided into four groups based on the data of blood lipids and WC. The prevalence of diabetes was the highest in the subgroup with abdominal obesity combined with dyslipidaemia. The difference in the prevalence amongst the four groups was statistically significant (χ^2^ = 45.16, *p* < 0.001).

As presented in Figure 5, the multivariate logistic regression analysis showed that the risk of diabetes was 2.28 times higher in the WC group and the dyslipidaemia group (OR = 2.28, 95% CI = 1.12–4.64, *p* = 0.023) than that in the group with normal WC/ortholiposis. The risk of diabetes in the subgroup with abdominal obesity combined with dyslipidaemia was 3.58 times higher (OR = 3.58, 95% CI = 2.49–5.13, *p* < 0.001). All these values were higher compared with those of the normal WC/ortholiposis group.

## 4. Discussion

The prevalence of diabetes has increased rapidly worldwide and has become one of the most serious public health problems. Wang et al. showed that diabetes has reached an epidemic proportion in China [8]. One in four people with diabetes worldwide come from China, where nearly 12% of adults have diabetes and over 50% have prediabetes [37]. Insulin resistance and β-cell dysfunction are considered the main causes of diabetes [38]. As the epidemic of diabetes continues unabated, further expanding the research on the prevention and control of diabetes is particularly important.

To the best of the authors’ knowledge, this study was the first to explore the combined effects of different types of obesity and dyslipidaemia on the prevalence of diabetes in middle-aged and elderly Chinese residents. Dyslipidaemia is a common biochemical abnormality in diabetes mellitus, and it is characterised by increased fasting and postprandial plasma TG concentration, decreased HDL-C levels and normal or elevated LDL-C concentration [39]. One study has shown that the ratio of TG to HDL-C concentration is related to insulin resistance, and an increase in the ratio indicates the aggravated secretion function of islet β-cells [40]. Another study has also shown that systemic obesity can lead to insulin resistance and promote abnormal blood glucose metabolism, which ultimately leads to diabetes [41].

This study revealed that patients with dyslipidaemia in all BMI categories were at risk of developing diabetes. Moreover, the risk of diabetes was significantly higher under a combination of a higher BMI and dyslipidaemia compared with the presence of dyslipidaemia or a high BMI alone. Therefore, the combined exposure to systemic obesity and dyslipidaemia can increase the risk of diabetes. On the one hand, in obesity, the insulin-mediated activation of the PI 3-kinase pathway is impaired, whereas the activation of the extracellular signal-regulated protein kinase 1/2 and the production of endothelin-1 by insulin are normal. This finding results in an impaired vasodilator, producing a net vasoconstriction response of insulin to muscular-resistant arteries [42]. As a consequence, vascular resistance increases, possibly contributing to the extensive periods of poor delivery of insulin, glucose and other nutrients to the muscle cells and impairing insulin-mediated glucose disposal in muscle, which is an important pathway for obesity’s contribution to insulin resistance [42]. On the other hand, studies have shown that cholesterol homeostasis is important for the adequate insulin secretory performance of β-cells, and lipid disorder is the promoting factor of insulin secretion defects and glucose metabolism changes. LDL-C and low HDL-C levels, as independent risk factors for β-cell dysfunction, increase the bioavailability of LDL-C in the pancreatic cell metabolism to have a cytotoxic effect; they may also increase β-cell apoptosis, with a direct effect of insulin resistance on diabetes [43]. Dyslipidaemia is caused by the change in lipoprotein activity, and it can further develop into insulin resistance and indirectly lead to diabetes [44].

In addition to directly affecting insulin resistance, in individuals with obesity, the increased flow of free fatty acids to the liver leads to the accumulation of liver TGs and increases the synthesis of LDL-C [44]. A study on a South Asian population showed that people with abdominal obesity have excess fat in their liver and skeletal muscles, thus increasing the risk of insulin resistance [45]. Previous studies have shown that the standardised rates of overweightness, general obesity and abdominal obesity in Chinese patients with diabetes were 44.4%, 41% and 45.4%, respectively [46]. Chinese people have a lower obesity rate than Westerners, and their body fat mainly accumulates in the abdomen [47].

The strength of this study was that the data were obtained from CHARLS, which is a nationwide population survey with a very high response rate. Therefore, it has advantages, such as detailed information on the demographic and potential confounders and the prospective study design. General conclusions were drawn, and the results can be considered representative of the whole population of Chinese middle-aged and elderly people. Compared with previous analyses, the present study analysed the combined effects of obesity and dyslipidaemia on the prevalence of diabetes in the population rather than the independent effects of obesity and dyslipidaemia on diabetes. Furthermore, the present study used a large sample size, with blood testing and physical measures, which provided relatively strong pieces of evidence. Various potential confounders, such as hypertension, dyslipidaemia and sleep duration, were also controlled to minimise bias. The results can provide a reference for the health management of patients with diabetes.

However, several limitations of this study should be considered. Firstly, this study was a cross-sectional survey research project. Longitudinal studies are needed to elucidate and verify the casual relationships amongst different types of obesity, dyslipidaemia and diabetes with respect to the combined effects of the two factors on diabetes. Secondly, no specific consideration was given to the specific categories of dyslipidaemia. Studies should further explore the combined effects of different lipid index abnormalities and obesity. Thirdly, certain data, such as sleep duration, were examined through questionnaires. Therefore, this study has inherent limitations in terms of the objectivity and reliability of certain data compared with objective measurements. Fourthly, certain potential confounding factors may not have been fully included in our analysis. Grip strength, for example, was excluded. In addition, abnormal O-GlcNAcylation expression levels were involved in the ethology of diabetes [48]. However, it was also excluded because the CHARLS database lacks the relevant data. These factors will be included in our future detailed investigations. Finally, the sample size of this study was not sufficiently large. Thus, large-scale studies should be carried out in the future to achieve more realistic results.

## 5. Conclusions

In conclusion, this study suggested that middle-aged and elderly people who have dyslipidaemia and systemic or abdominal obesity are more likely to develop diabetes. The results also implied that monitoring and controlling lipid levels in people with obesity are effective methods to prevent diabetes. The findings are expected to be useful as empirical data for the establishment of strategies to prevent diabetes amongst middle-aged and elderly people. In addition, the screening of at-risk populations and the early detection and monitoring of blood sugar levels are key elements in proper prevention and therapy. Currently used diagnostic tools suffer from specific limitations. Consequently, several cases remain undiagnosed, and patients already exhibit complications at the time of the first diagnosis. Therefore, new and more sensitive biomarkers should be discovered for the detection of pre-diabetic conditions.

## Figures and Tables

**Figure 1 ijerph-19-08036-f001:**
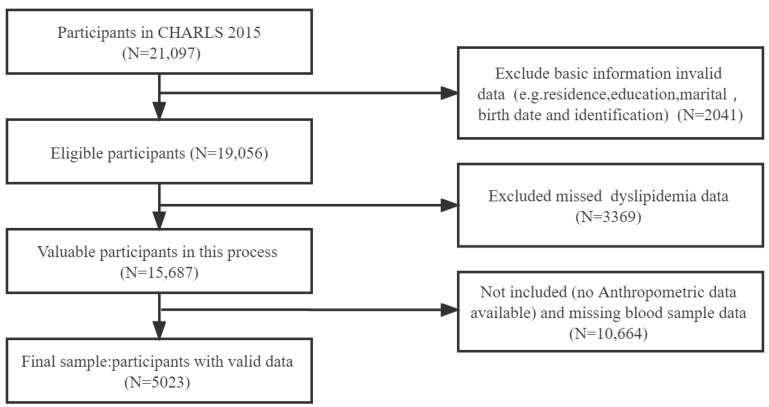
Flow diagram of the population in the project.

**Figure 2 ijerph-19-08036-f002:**
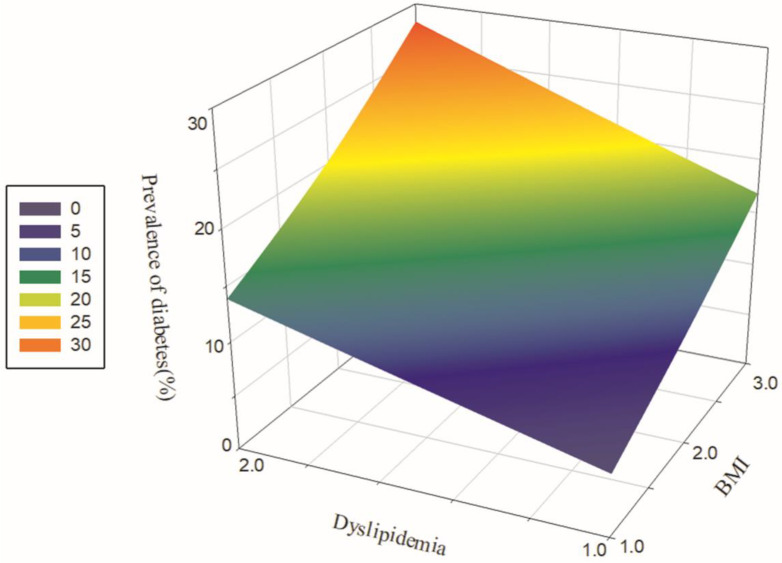
Relationship amongst dyslipidaemia, BMI and the prevalence of diabetes. Note: BMI: 1 = < 24 kg/m^2^, 2 = 24–27.9 kg/m^2^, 3 = ≥ 28 kg/m^2^; dyslipidaemia: 1 = no, 2 = yes.

**Figure 3 ijerph-19-08036-f003:**
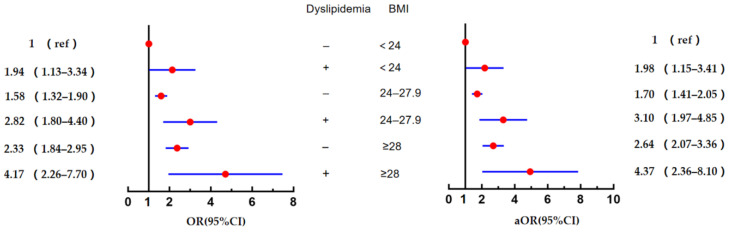
Combined effects of systemic obesity and dyslipidaemia on diabetes. Note: Data presented with outcome, OR and 95% CI; aOR, adjusted odd ratio (adjusted variables were age, smoking status and hypertension).

**Figure 4 ijerph-19-08036-f004:**
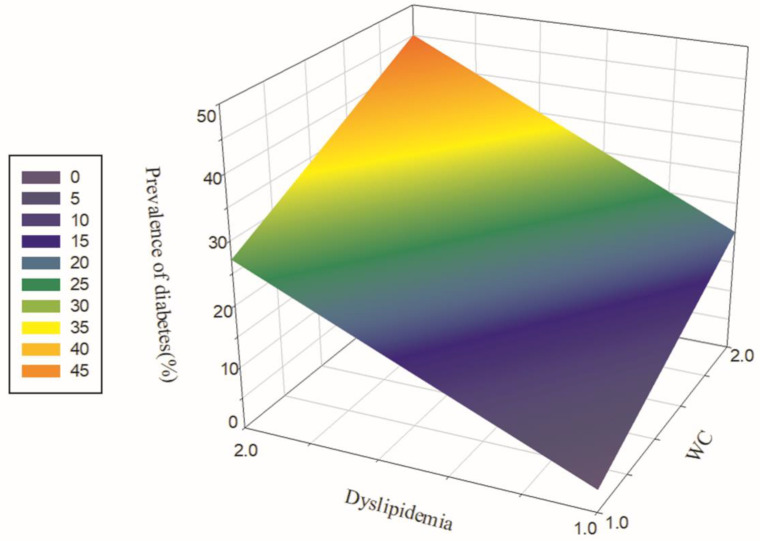
Relationship amongst dyslipidaemia, abdominal obesity and the prevalence of diabetes. Note: WC: 1 = normal, 2 = abdominal obesity; dyslipidaemia: 1 = no, 2 = yes.

**Figure 5 ijerph-19-08036-f005:**
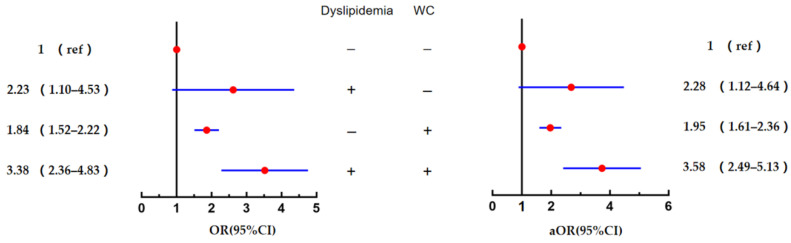
Combined effects of abdominal obesity and dyslipidaemia on diabetes. Note: Data presented with outcome, OR and 95% CI; aOR, adjusted odd ratio (adjusted variables were age, smoking status and hypertension).

**Table 1 ijerph-19-08036-t001:** The prevalence of diabetes in people with different characteristics.

Variables	n (%)	Diabetes		χ^2^	*p*
		Yes (%)	No (%)		
Gender				1.73	0.188
Male	1202 (23.93)	185 (25.87)	1017 (23.61)		
Female	3821 (76.07)	530 (74.13)	3291 (76.39)		
Age (years)				14.99	0.001
≤60	3167 (63.05)	411 (57.48)	2756 (63.97)		
61–70	1198 (23.85)	182 (25.45)	1016 (23.58)		
≥70	658 (13.10)	122 (17.06)	536 (12.44)		
Residence				1.30	0.254
Urban	758 (15.63)	118 (16.50)	640 (14.86)		
Rural	4265 (84.91)	597 (83.50)	3668 (85.14)		
Hypertension				18.23	<0.001
Yes	4586 (91.30)	623 (87.13)	3963 (91.99)		
No	437 (8.70)	92 (12.87)	345 (8.01)		
Nap duration (minutes)				5.77	0.056
0	2292 (45.63)	302 (42.24)	1990 (46.19)		
0–30	325 (6.47)	41 (5.73)	284 (5.73)		
≥30	2406 (47.90)	372 (52.03)	2034 (52.03)		
Drinking frequency				1.64	0.441
≥1/month	898 (17.88)	131 (18.32)	767 (17.80)		
≤1/month	441 (8.78)	71 (9.93)	370 (8.59)		
Never	3684 (73.34)	513 (71.75)	3171 (73.61)		
Marital status				3.66	0.056
Married	4428 (88.15)	615 (86.01)	3813 (88.51)		
Unmarried/divorced/widowed	595 (11.85)	100 (13.99)	495 (11.49)		
Education level				2.34	0.311
Illiterate	4517 (89.93)	654 (91.47)	3863 (89.67)		
Primary school/Middle school	422 (8.40)	52 (7.27)	370 (8.59)		
High school and above	84 (1.67)	9 (1.26)	75 (1.74)		
Dyslipidemia				22.61	<0.001
Yes	4774 (95.04)	654 (91.47)	4120 (95.64)		
No	249 (4.96)	61 (8.53)	188 (4.36)		
Night sleep duration (h)				0.28	0.594
≤5	1525 (30.36)	211 (29.51)	1314 (30.50)		
>5	3498 (69.64)	504 (70.49)	2994 (69.50)		
Smoking status				6.10	0.014
Yes	573 (11.41)	101 (14.13)	472 (10.96)		
No	4450 (88.59)	614 (85.87)	3836 (89.04)		
Abdominal obesity				45.16	<0.001
Yes	1759 (35.02)	171 (23.92)	1588 (36.86)		
No	3264 (64.98)	544 (76.08)	2720 (63.14)		
BMI (kg/m^2^)				66.74	<0.001
<24	2744 (54.63)	301 (42.10)	2443 (56.71)		
24–27.9	1676 (33.37)	277 (38.74)	1399 (32.47)		
≥28	603 (12.00)	137 (19.16)	466 (10.82)		

**Table 2 ijerph-19-08036-t002:** Relationship amongst systemic obesity, abdominal obesity, dyslipidaemia and diabetes.

Variables	Model 1		Model 2	
	OR (95% CI)	*p*	OR (95% CI)	*p*
Systemic obesity ^a^				
Yes	1.79 (1.53–2.10)	<0.001	1.56 (1.28–1.89)	<0.001
No	1 (ref)		1 (ref)	
Abdominal obesity ^b^				
Yes	1.86 (1.55–2.23)	<0.001	1.92 (1.60–2.31)	<0.001
No	1 (ref)		1 (ref)	
Dyslipidemia ^c^				
Yes	2.04 (1.51–2.76)	<0.001	1.82 (1.34–2.47)	<0.001
No	1 (ref)		1 (ref)	

Note: ^a^ Adjusted for age, hypertension, smoking status, abdominal obesity and dyslipidaemia; ^b^ adjusted for age, hypertension, smoking status, systemic obesity and dyslipidaemia; ^c^ adjusted for age, hypertension, smoking status, systemic obesity and abdominal obesity.

## Data Availability

The CHARLS data are publicly available at http://charls.pku.edu.cn.

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
