# Peer review of "Combined Effects of Obesity and Dyslipidaemia on the Prevalence of Diabetes Amongst Adults Aged ≥45 Years: Evidence from a Nationally Representative Cross-Sectional Study"

_ijerph, 2022, doi:10.3390/ijerph19138036_

Round 1
Reviewer 1 Report
1-The study had to be conducted on a large number in order for the study to have more realistic results
2- Reference No. 22 and 23 is very old and needs to be updated
Reviewer 2 Report
This manuscript used the data of the China Health and Retirement Longitudinal Study (CHARLS) in 2015 to analyze the relationship amongst BMI, WC, dyslipidemia and the prevalence of diabetes in middle-aged and elderly Chinese population and provide a reference for the formulation of primary prevention strategies for diabetes. The scientific collect is very interesting, however, some aspects, as indicated below, should be addressed before the document can be considered for publication in this journal. This version of the manuscript is not enough complete.
Minor revision:
-English language and style are fine/minor spell check required.
-I suggest to review the style of the manuscript according to the guidelines of the journal.
Major revision:
Line 54. Diabetes mellitus type 2 is a metabolic disease characterized by hyperglycemia, consisting of high glucose levels chronically present in the blood. Oxidative stress is critically involved in diabetes pathogenesis. Red cells are constantly exposed to glucose in the blood, and they have been widely investigated for their important role in different physiological conditions due to the fact of their sensitivity to oxidative stress. Among the consequences of hyperglycemia, glycation of proteins has also been considered. In this regard, a typical example is provided by glycated hemoglobin (A1c), considered as a standard test to monitor glycemic status since. In this regards, (line 56) I suggest to add the following reference (DOI: 10.3390/antiox9050365), in order to discuss also this important aspect (Link between diabetes, hyperglycaemia and oxidative stress).
Line 56. Dyslipidemia is also related to many factors, including poor blood glucose control, insulin resistance, inflammation and genetic susceptibility [21]. Observations confirmed the putative utility of dyslipidemia in CVD risk prediction in individuals with diabetes [22]. Dyslipidemia is also related to many factors, including poor blood glucose control, insulin resistance, inflammation and genetic susceptibility [21]. Observations confirmed the putative utility of dyslipidemia in CVD risk prediction in individuals with diabetes [22]. In this regard, I suggest to add this recent reference (DOI: 10.1007/s11886-021-01455-w).
-It is widely established that poorly controlled or uncontrolled diabetes leads to hyperglycemia, and consequently, to increased O-glycosylation (O-GlcNAc levels) in various tissues. O-Glycosylation is the post-translational conjugation of a single monosaccharide, N-acetylglucosamine, to Serine or Threonine residues of nuclear and cytoplasmic proteins. O-GlcNAcylation is catalyzed by the enzymes O-GlcNAc-transferase (OGT) and O-GlcNAcase (OGA). In fact, human erythrocyte proteins are highly O-GlcNAcylated, and O-GlcNAc of erythrocyte proteins is regulated in response to glycemic status. Interestingly, different groups found that OGA expression was increased in erythrocytes from individuals with both pre-diabetes and diabetes compared to the control population, and therefore suggested that OGA levels could be even superior to HbA1c in detecting a condition of pre-diabetes. In addition, genetic variation in O-GlcNAc regulatory enzymes could play a role in the genetic predisposition to T2D.
Has been this aspect in this population considered and/or analyzed?
Could authors add these data, if there are, in the following collect?
-Screening of at-risk populations, early detection and monitoring of blood sugar levels are key elements in proper prevention and therapy. Currently used diagnostics tools suffer of limitations. Consequently, several cases remain undiagnosed and patients already exhibit complications at first diagnosis. Therefore, there is a need of new and more sensitive biomarkers for detecting the pre-diabetic condition.
Based on this considerations, I suggest to enhance the conclusions.
Reviewer 3 Report
Dear Authors,
Congratulation for your interesting article, it is well justified, the methodology and the results are clearly exposed. We have only few suggestions to improve the article.
In your abstract you write: “Residents with self-reported diagnoses of diabetes were selected as subjects and a total of 6233 valid ones were obtained.”, on the basis of this information we suppose that self-reporting diagnoses of diabetes is an inclusion criterium. In table 1, we can see that from those 6233 peoples only 164 had effectively a diagnostic of diabetes. It is a surprising result! If the self-reported diagnose of diabetes was an inclusion criterium, it must have a reason, probably studying significative population of people with diabetes and in this case, it should be better to select the people on the basis of their health and physical measurements. Looking at figure 1, we can interpret that what you really did is selecting in the participants in Charls, the ones that offered completes data’s (basics, anthropometrics and blood information’s), from those subjects 164 had diagnoses of diabetes, and in this case self-reported diagnoses is without interest. Could you please clarify this and let the future lecturers better understand what you did?
In CHARLs data basis you have information’s about physical activity, including hand grip measurement, the prevention of physical activity on diabetes is well recognised in the scientifically literature. Do you have consistent reasons for not using it in the adjusting factors? If yes it should be important to justify that, if not, it should be discussed as a limit of your study.
We wish you good success in the publication of your article.
Round 2
Reviewer 2 Report
The authors improved the manuscript, thus incorporating most of the suggestions.
Thus, this manuscript can be published in the present form.